# Statins Aggravate the Risk of Insulin Resistance in Human Muscle

**DOI:** 10.3390/ijms23042398

**Published:** 2022-02-21

**Authors:** Stefanie A. Grunwald, Stefanie Haafke, Ulrike Grieben, Ursula Kassner, Elisabeth Steinhagen-Thiessen, Simone Spuler

**Affiliations:** 1Muscle Research Unit, Experimental and Clinical Research Center, Max Delbrück Center for Molecular Medicine in the Helmholtz Association (MDC), Charité–Universitätsmedizin Berlin, Corporate Member of Freie Universität Berlin and Humboldt-Universität zu Berlin, 13125 Berlin, Germany; stefanie.haafke@charite.de (S.H.); ulrike.grieben@t-online.de (U.G.); 2Interdisciplinary Lipid Metabolic Center, Charité Universitätsmedizin Berlin, 13353 Berlin, Germany; ursula.kassner@charite.de (U.K.); elisabeth.steinhagen-thiessen@charite.de (E.S.-T.)

**Keywords:** statins, insulin resistance, AMPK, AKT, human skeletal muscle, primary human muscle cells

## Abstract

Beside their beneficial effects on cardiovascular events, statins are thought to contribute to insulin resistance and type-2 diabetes. It is not known whether these effects are long-term events from statin-treatment or already triggered with the first statin-intake. Skeletal muscle is considered the main site for insulin-stimulated glucose uptake and therefore, a primary target for insulin resistance in the human body. We analyzed localization and expression of proteins related to GLUT4 mediated glucose uptake via AMPKα or AKT in human skeletal muscle tissue from patients with statin-intake >6 months and in primary human myotubes after 96 h statin treatment. The ratio for AMPKα activity significantly increased in human skeletal muscle cells treated with statins for long- and short-term. Furthermore, the insulin-stimulated counterpart, AKT, significantly decreased in activity and protein level, while GSK3ß and mTOR protein expression reduced in statin-treated primary human myotubes, only. However, GLUT4 was normally distributed whereas CAV3 was internalized from plasma membrane around the nucleus in statin-treated primary human myotubes. Statin-treatment activates AMPKα-dependent glucose uptake and remains active after long-term statin treatment. Permanent blocking of its insulin-dependent counterpart AKT activation may lead to metabolic inflexibility and insulin resistance in the long run and may be a direct consequence of statin-treatment.

## 1. Introduction

Statins, the most widely used cholesterol-reducing drugs worldwide, have been associated with type 2 diabetes mellitus (T2DM). The beneficial effect of statins reducing the risk of cardiovascular disease is beyond question. Statins inhibit the 3-hydroxy-3-methyl-glutaryl coenzyme A reductase (HMGCR), the rate-limiting enzyme in cholesterol biosynthesis reducing blood cholesterol levels. Side effects include statin-associated muscle symptoms, i.e., myalgia and elevated creatine kinase levels, proteinuria, and elevated liver enzymes. [1] There has been solid and consistent evidence in the last decade that statins provoke insulin resistance, increasing the risk for T2DM [2,3,4,5]. Depending on statin type, dosage, and a combination of individual risk factors, i.e., age, BMI, pre-metabolic syndrome, statins may increase the risk of T2DM by up to 25% [6,7,8,9,10,11,12]. Although several studies have aimed to link statin-induced insulin resistance to pathophysiological changes in humans, the molecular mechanism is not understood. Most of the data are obtained from studies in hepatic, endothelial, and pancreatic cells in vitro or animal models and only very few in skeletal muscle cells. The skeletal muscle is the largest organ for insulin-stimulated glucose uptake with about 80% of body total glucose disposal [13,14]. Glucose uptake into skeletal muscle is mediated by a relocation of the glucose transporter GLUT4 from intracellular vesicles to cholesterol-rich membrane compartments, i.e., caveolae. GLUT4 translocates via insulin-signaling protein kinase B/Akt activation or via muscle contraction induced 5′-adenosine monophosphate–activated protein kinase (AMPK) phosphorylation. In patients with insulin resistance and T2DM, GLUT4 translocation to the membrane is disturbed and glucose uptake is reduced in skeletal muscle [15,16,17,18]. Sanvee et al. and Seshadri et al. report on impaired insulin signaling, reduced glucose uptake, and higher amounts of fatty acids after chronic and short-term statin treatment in vitro and in vivo in rodents [19,20]. Simvastatin, atorvastatin, and rosuvastatin have been linked to insulin resistance and increased T2DM prevalence in statin patients [21,22,23]. A phase 4 clinical trial completed in 2020 examined the relationship between insulin resistance and statin-induced T2DM using high-dose atorvastatin and was able to show increased insulin resistance and insulin secretion in the high-dose atorvastatin group [24]. Individuals with preexisting insulin resistance and a risk of developing diabetes may be at increased risk of developing T2DM with statin treatment [25]. In primary human muscle cells, we showed that two statins, simvastatin and rosuvastatin, alter fatty-acid metabolism as well as glucose related events. [26] The two changes may be related. Taken together, the data suggest that an increased risk for insulin resistance and T2DM is causally linked to statin use, through a yet unknown molecular mechanism.

The present study addresses the still unknown cellular mechanisms of increased insulin resistance during long-term statin therapy in patients and short-term statin administration in vitro. Patients ingest statins for many years before insulin resistance becomes evident. We wondered whether the molecular changes in skeletal muscle glucose metabolism in statin patients is a long-term effect or is detectable shortly after first treatment and whether data reported from non-human muscle are translatable to the human situation. We hypothesize that alterations in insulin signaling by statins in skeletal muscle are early events and can be detected shortly after first statin-treatment initiation in human muscle cells in vitro. Primary human skeletal muscle cells are an optimal model for studying metabolic events and signal transduction under defined conditions, controlling for any influence on glucose and fatty acid metabolism, i.e., glucose uptake. We tested the effect of the most common lipophilic (simvastatin) and a hydrophilic (rosuvastatin) statin on glucose uptake in human primary skeletal muscle cells obtained from statin-naïve patients and one statin-treated patient without muscle side effects. We further analyzed proteins involved in glucose metabolism in skeletal muscle biopsy specimens from long-term statin-treated patients without a T2DM diagnosis. Statin treatment leads to increased pAMPKα levels after short-term treatment that can be also detected in skeletal muscle tissue from patients. Furthermore, GLUT4 is normally distributed whereas CAV3 tends so localize more intracellular in statin-treated primary muscle cells.

## 2. Results

We studied the impact of statins on glucose metabolism at the protein level in human muscle. Characteristics of our study subjects are presented in Table 1.

### 2.1. Long-Term Statin-Treatment Activates AMPKα in Human Skeletal Muscle

Glucose is transported into skeletal muscle cells by GLUT4 translocation to the plasma membrane and T-tubules. Additionally, GLUT4 levels at the plasma membrane increase from 5% to about 50% through stimulation compared to basal conditions. Muscle tissues from patients treated with statins until the day of biopsy and statin naïve controls were analyzed for GLUT4 localization (Figure 1a).

In all samples analyzed, GLUT4 was normally distributed and found at the plasma membrane, close to the nucleus, and near the T-tubules. The GLUT4 protein expression levels determined by Western blot were not statistically significantly different between statin patients and controls (Figure 1b,c).

The translocation of GLUT4 is mediated either by AKT via insulin stimulation or AMPK activation via contraction. The AKT/mTOR pathway is essential for skeletal-muscle cell regeneration. We further analyzed AMPKα, phosphorylated AMPKα, AKT, phosphorylated AKT, GSK3ß, and mTOR which are known to be involved in glucose metabolism (Figure 1b,c). As is known, AMPKα is phosphorylated and activated by several factors, i.e., insulin, leptin, calcium changes, or muscle contraction. Total AMPKα levels were not differentially expressed between statin patients and controls, but we found a statistically significant increase in phosphorylated AMPKα in statin-treated patients of 77% (Figure 1b–d). Phosphorylated AKT blocks AMPKα activity. The total AKT levels were significantly higher in statin-treated patients, whereas phosphorylated AKT levels were lower and consequently 22% less at activated state (Figure 1b–d). Activated AKT inhibits GSK3ß by phosphorylation, whereas mTOR activates GSK3ß, which is necessary for glycogen synthesis. The mTOR levels were slightly increased and GSK3ß levels were not different in statin-treated patients compared with controls (Figure 1b,c).

### 2.2. Short-Term Statin Treatment in Human Primary Myotubes Affects Glucose Metabolism via AMPKα Similar to the Situation in Human Skeletal Muscle

We further tested whether the effect on glucose metabolism found in human skeletal muscle tissue could be detected in vitro in human primary myotubes under statin treatment for 96 h w/o stimulation with glucose and insulin (Figure 2 and Figure 3). Normally, GLUT4 is distributed in vesicular structures around the nucleus and at the plasma membrane (Figure 2, arrows). Furthermore, GLUT4 signals in simvastatin-treated samples appear less punctuated, which was expected for vesicular structures and observed for controls. The GLUT4 protein expression levels were not statistically significantly different between statin treatment and controls (Figure 3a,c–i). GLUT4 is recruited to lipid rafts at the plasma membrane, i.e., caveolae, dependent on membrane cholesterol levels. We analyzed caveolin 3 (CAV3) localization, a structural protein of caveolae in skeletal muscle cells. CAV3 accumulates around the nucleus and is less found at the plasma membrane after simvastatin and to some extent after rosuvastatin treatment (Figure 2, arrows).

In non-stimulated, statin-treated human primary myotubes, AMPKα levels were decreased with both statins, but active AMPKα remained at similar expression levels in all groups (Figure 3a,c–i). Stimulation with glucose and insulin slightly increased these protein levels, but they remained significantly decreased in simvastatin-treated myotubes. A change in the AMPKα total content confirms that changes in AMPKα phosphorylation levels between the groups and a ratio may be favored. The pAMPKα/AMPKα ratio increased by 139% and 76% for simvastatin and rosuvastatin, respectively, in non-stimulated human primary myotubes and decreased within the stimulated group (Figure 3a). In contrast, we found normal AKT levels in all groups, whereas phosphorylated AKT was decreased under simvastatin but unchanged with rosuvastatin without stimulation (Figure 3a,c–i). These levels were more diminished in stimulated statin-treated myotubes. The pAKT/AKT ratio decreased by 65% for simvastatin and increased by 92% for rosuvastatin in non-stimulated human primary myotubes but decreased further within the stimulated group (Figure 3a). The GSK3ß levels (Figure 3a,c–i), inhibited by phosphorylated AKT, as well as those of mTOR (Figure 3a,c–i), which is relevant for GSK3ß activation, were both decreased under simvastatin. mTOR levels also decreased after treatment with rosuvastatin in the non-stimulated group. Unexpectedly, mTOR protein levels increased within 5 min of insulin/glucose stimulation. We therefore tested for mTOR activity regulated by phosphorylation. mTOR phosphorylation (Ser2448) was not significantly different from the unstimulated control under any condition (Figure 3b). Although mTOR total and phosphorylation levels were determined using different methods, decreased total mTOR levels after statin treatment and normal mTOR phosphorylation (Ser2448) compared to control indicate an increased mTOR phosphorylation (Ser2448) ratio for statin-treated primary human myotubes.

## 3. Discussion

Statin-induced pathomechanisms are a significant clinical and economical challenge when treating patients at high risk of cardiological events. It is becoming increasingly evident from clinical studies that diabetes is a risk factor in statin-treated patients [7,11]. Although insulin resistance and impaired glucose uptake has been observed in liver, ß-cells, and skeletal muscle from rodents, contradictory data obtained from rats showed stimulated insulin/AKT pathway activation through statins [27]. Thus, there is a need for studies in human skeletal muscle, the main glucose consumer. We recently reported that statin treatment interfered via HMG-CoA accumulation into acetyl CoA-dependent lipid and fatty acid metabolism [26]. Lipid accumulation and changes in free fatty-acid-mediated pathways have been linked to insulin resistance in skeletal muscle from statin patients [28,29] and L6 myotubes [22]. Insulin resistance in skeletal muscle is characterized by metabolic inflexibility as it fails to switch between the two main substrates as energy resource – glucose and fatty acids [30]. Furthermore, AMPK is a central protein in the regulation of GLUT4-mediated glucose uptake by muscle contraction, and fatty acid oxidation by sensing cellular ATP and calcium levels. Both insulin stimulation via AKT and muscle contraction via AMPK regulate GLUT4 translocation and GLUT4 expression, thereby modulating glucose homeostasis in skeletal muscle. A significant increase in AMPKα activation by phosphorylation at Thr172 is evident in primary human myotubes under statin treatment and can also be detected in skeletal-muscle tissue from patients treated with statins for long-term. It suggests low cellular energy levels and/or changes in calcium levels favoring catabolic, i.e., fatty acid oxidation, while inhibiting anabolic pathways, i.e., lipogenesis and sterol synthesis. Interestingly, active AMPK can inactivate HMGCR via phosphorylation [31] and may amplify a statin cholesterol reducing effect through acetyl-CoA and fatty acid accumulation that again may further activate AMPKα.

Reduced membrane cholesterol has been described as a mechanism of AMPK action, directly linked to GLUT4 trafficking [32]. Active AMPKα mediates glucose transport into skeletal muscle cells by GLUT4 translocation to the cholesterol-rich plasma membrane compartments, i.e., caveolae, and T-tubules. In contrast to data from C2C12 myotubes, 3T3L-1 mouse adipocytes, and rodents [19,33], GLUT4 distribution and GLUT4 protein levels were maintained in statin-treated patients and primary human muscle cells. Furthermore, GLUT4 has been shown to translocate to caveolae in adipocyte [34,35] and the caveolae protein CAV3 attracts GLUT4 localization to the plasma membrane in muscle cells [36]. Moreover, CAV3 accumulates in statin-treated primary human myotubes without significant changes to GLUT4 localization as it has been shown for GLUT4 and caveolin-1 by atorvastatin leading to impaired glucose uptake and insulin resistance in muscle, adipose tissue, and liver [16,17,37]. Changes in plasma-membrane cholesterol levels by statins and plasma membrane integrity may cause this CAV3 intracellular accumulation. As stated in the literature, under insulin stimulation, AKT is activated, blocks AMPKα activity, and initiates GLUT4 translocation to the plasma membrane for glucose uptake into the cell [38,39]. AKT phosphorylation is impaired under statin treatment in both, primary human muscle cells and skeletal muscle tissues from patients, as it has been found in C2C12 myotubes and rodents [19,20,40]. Interestingly, Seshadri et al. linked decreased AKT phosphorylation in L6 myotubes to fatty-acid accumulation, as previously described by us [26]. The mechanism by which statins interfere with AKT activation may be related to AMPK activation and impaired protein prenylation through HMGCR inhibition, which is important for AKT activity [41,42]. As we did find decreased activation of AKT, AMPK activity was increased, leading to GLUT4 translocation to cell membrane and GLUT4 expression levels comparable to controls. Another study performed by Zhao and Zhao (2015) also showed decreased p-AKT but contrary to our results found GLUT4 expression reduced and p-p38 MAPK expression was reduced, which is downstream of AMPK and explains the decrease in GLUT4 expression found. Diversities in the donor and statin concentration used may explain the difference between the studies. However, activated AMPK increased and thus GLUT4 expression did not significantly decreased in our study, which includes results from 20 different donors with individual variances being found. GLUT4 expression levels were confirmed at protein level by immunohistochemistry and Western blot in tissues from statin patients and in the statin-treated muscle cells [43,44].

Supporting our data, we found that LANCL2, a remarkable AKT activator [45], decreased by 50% in our proteome data from statin-treated human myotubes [26]. It suggests that in statin-treated muscle, GLUT4 translocates to plasma-membrane compartments through AMPKα activation rather than insulin-induced AKT activation. Insulin resistance is a likely a result. Interestingly, the phase 4 clinical trial published by Abbasi et al. in 2021 presents insulin resistance in high-dose atorvastatin-treated patients associated with changes in insulin secretion. Insulin secretion is increased temporarily in these patients, presumably to compensate for the onset of insulin resistance in the short term [24]. However, if insulin resistance persists, insulin secretion appears to decrease again and insulin needs cannot be supplied. This study confirms our data and may explain the differences we observed between short-term statin treatment and long-term statin therapy. Decreased insulin secretion appears to be present in many in vivo studies and in vitro studies with statins as reviewed by Carmena and Betteridge 2019 [46]. However, it is not clear whether it is a primary or secondary effect, but the associated molecular mechanisms occur early in skeletal muscle.

In different cell types, such as adipocytes and ß-islet cells, statins are described to have a positive effect on insulin sensitivity and hyperglycemia. However, this is different for muscle, which may switch to an alternative pathway for GLUT4 translocation [43]. Statins inhibit the formation of isoprenoids and thus farnesyl and geranylgeranyl pyrophosphates, from the mevalonate pathway and required for small GTPase prenylation, are reduced [43,47]. RabGTPases are hydrophobically prenylated by Rab geranylgeranyl transferase to bind to their target membrane structures, such as GLUT4 vesicles, and to direct the translocation of GLUT4 [48,49]. Furthermore, the prenylation of Ras and Rab leads to the phosphorylation of PI3K, which activates Akt and mTOR, which is decreased in our study. However, the activation of AKT leads to the phosphorylation and inactivation of TBC1D1 and TBC1D4. The activity of TBC1D1 and TBC1D4 is thought to inhibit the activity of Rab GTPase-activating protein toward certain Rab isoforms. Inactivation of TBC1D1 and TBC1D4 results in the increased generation of more active GTP-loaded Rab. Active GTP-loaded Rab promotes GLUT4 translocation to the plasma membrane. Active AMPK can also mediate the emergence of more active GTP-loaded Rab [43,47,48]. The prenylation of small GTPases may be depleted over time via the inhibition of the mevalonate pathway and even by an increase in HMGCR expression, as we showed recently by Grunwald et al. in 2020 under statin treatment [26]. The results from muscle-tissue samples from patients treated with statins for years show normal GLUT4 translocation. None of our patients had type 2 diabetes. It would be interesting to study changes in patients developing type 2 diabetes. Furthermore, AMPKα activation leads to mTOR inhibition via TSC complex 2 protein activation. Impaired mTOR signaling is involved in diabetes and cardiovascular events [50]. After short-term statin treatment, we were able to demonstrate that decreased mTOR levels were associated with a decrease in AKT activation, which may disturb the physiological response to insulin in statin-treated human primary myotubes. Significant changes in mTOR phosphorylation, which could explain these results, were not evident. Active AKT phosphorylates and negatively regulates GSK3ß, which is involved in cellular processes such as cell survival and proliferation but also insulin-stimulated glycogen synthesis in muscle. Decreased GSK3ß levels were only evident in short-term simvastatin-treated primary human myotubes. The role of GSK3ß in statin-induced pathomechanisms in skeletal-muscle cells needs further investigation. 

Our study provides valuable insight to elucidate the molecular mechanisms associated with insulin resistance under statin treatment. We show that short-term statin treatment affects AMPKα and AKT activity in human skeletal-muscle primary myotubes that remains detectable in skeletal muscle biopsy specimen from statin-treated patients. We found differences to the data obtained from non-human or non-muscle model systems in the literature. It is questionable whether these differences are dependent on the duration of statin treatment or the model used. Muscle cells are a purified cellular model under defined experimental conditions, whereas skeletal-muscle-tissue sections consist of different cell types. We suggest that permanent AMPKα activation through statins and the blocking of its insulin-dependent counterpart AKT may lead to metabolic inflexibility and insulin resistance in the long run. This may be a direct consequence of HMGCR inhibition and a self-amplifying effect. 

## 4. Materials and Methods

We analyzed glucose metabolism at the protein level in primary human muscle cells from twelve donors and skeletal muscle tissues from eight statin patients. No relevant gender-specific difference was found in the results.

### 4.1. Human Primary Myoblasts

Muscle biopsy specimens for myoblast isolation and analyses at the tissue level were obtained from M. vastus lateralis and M. triceps brachii after IRB approval by the regulatory agencies (EA1/203/08 and EA2/051/10, Charité Universitätsmedizin Berlin, in compliance with the Declaration of Helsinki) at the HELIOS Hospital Berlin Buch, Berlin, Germany, for diagnostic or orthopedic reasons. All study participants provided informed consent. Human muscle biopsy material is restricted due to availability and quantity and for ethical reasons. Muscle biopsy handling was performed as described previously [51]. Statin-myopathy patients had myalgias (≥2 of 6 on a visual analogue scale) and/ or creatine kinase levels elevated > 300 IU/l under therapy (Table 1). Myoblast isolation was performed as described previously [52]. The contamination of myoblast cultures with fibroblasts was always below 5% as assessed by anti-desmin staining.

We controlled for confounding factors including cell culture, sample handling, and biopsy-stress variations. At least two cell populations were used simultaneously in an experimental setup in the same experimenter. In an experimental setup, controls and both statins were completed simultaneously.

### 4.2. Reagents, Buffers, and Antibodies

The active forms of simvastatin (hydrophobic) and rosuvastatin (hydrophilic) were purchased from Calbiochem and LKT Laboratories. Both statins were dissolved in DMSO (Sigma Aldrich, St. Louis, MO, USA) and used at 5 µM concentrations after initial testing as recently published [26]. A skeletal-muscle growth medium was obtained from ProVitro (Berlin, Germany). Opti-MEM and GlutaMax were purchased from Invitrogen (Berlin, Germany). All antibodies are listed in Table 2. The pre-stained protein marker PageRuler™ Plus was from Thermo Scientific (St. Louis, MO, USA), and HiMark™ was obtained from Invitrogen (Berlin, Germany). Krebs Ringer buffer solution at pH 7.4 was composed of 25 mM HEPES, 135 mM NaCl, 3.6 mM KCl, 1 mM CaCl_2_, 0.5 mM MgSO_4_, 0.5 mM KH_2_PO_4_, and 5 mM NaOH. Protein lysis buffer at pH 7.4 contained 20 mM Tris, 150 mM NaCl, 1 mM EDTA, 1 mM EGTA, 1% TritonX-100, 1 mM PMSF, 1 mM glycerol phosphate, 1 mM Na_3_VO_4_, 1 mM NaF, 2.5 mM Na_4_O_7_P_2_, 1x PhosSTOP (Roche, Mannheim, Germany), and 1x Complete EDTA-free (Roche, Mannheim, Germany).

### 4.3. Glucose Uptake in Primary Muscle Cell Populations

Statin-treatment experiments were conducted with a total of seven different human primary myoblast cultures (Table 1) as described previously [26]. Primary human myoblasts were cultured until 70–80% confluence was reached and switched to differentiation in OptiMEM under statin-treatment (Invitrogen, Germany). An experimental sample set was untreated and treated with 7 mM DMSO or 5 µM statins. The medium was changed every 24 h to refresh the medium and supplements. After about 90 h, two sample sets per human myotube cell line were washed and starved in Krebs Ringer solution for 5 h including statins. One sample set was left non-stimulated and the other was stimulated with 480 nM insulin and 5 mM glucose for 5 min. All treatments were performed in parallel for each cell population at the same time of day to exclude circadian rhythm effects.

### 4.4. Immunofluorescence

Muscle cryosection GLUT4 immunostaining was performed as previously described [47]. Briefly, after fixation and permeabilization, sections were stained with rabbit GLUT4 antiserum followed by incubation with biotin-conjugated Fab2 and streptavidin-Cy3 (Table 2). Primary human muscle cells were fixed with 3.7% PFA and permeabilized with 0.2% Triton X-100. After blocking with 3% BSA in PBS, cells were incubated with CAV3 and GLUT4 (Table 2). Nuclei were counterstained with HOECHST 33258. Confocal images were obtained using a Zeiss LSM 700 (Jena, Germany). The investigator was blinded for each patient and sample group while capturing microscopic pictures from human muscle cryosection immunostainings.

### 4.5. Protein Expression Using Immunoblot and Sandwich ELISA

Cryosections (~250 μm) from skeletal muscle and primary human myotubes, pre-washed with 1 × ice-cold PBS, were solubilized in ice-cold protein lysis buffer. Proteins were separated on 8–16% Tris-glycine gradient gels and electrophoretically transferred to nitrocellulose membranes (Whatman, Dassel, Germany). Membranes were incubated with primary antibody overnight (Table 2). Protein bands were determined using an infra-red scanner (LI-COR, Lincoln, NE, USA) and ECL (SuperSignal West Dura ECL, Thermo Scientific, Rockford, MI, USA; ChemiSmart 5000, VILBER LOURMAT, Collégien, France). Quantitation was performed using Image Studio Lite ver5.2. Endogenous levels of mTOR protein phosphorylated at Ser2448 were analyzed using PathScan^®^ Phospho-mTOR Sandwich ELISA Kit (Cell Signaling Technology, Beverly, MA, USA) according to the manufacturer protocol. Because 16 out of 18 proteins samples from cryosections did not lead to valid signal intensity above threshold in ELISA, we excluded results from skeletal muscle tissue.

### 4.6. Statistics

Statistical analyses were performed with GraphPad Prism (v 8.0). We tested for normality using the Shapiro–Wilk test (alpha = 0.05). For muscle-tissue Western blotting results, we applied the Mann–Whitney test. One-way ANOVA was corrected for multiple comparisons (Dunn’s test) for primary human muscle cells data. Values of p are shown by asterisks using the standard convention: * *p* ≤ 0.05; ** *p* ≤ 0.05; *** *p* ≤ 0.005; **** *p* ≤ 0.0005.

## Figures and Tables

**Figure 1 ijms-23-02398-f001:**
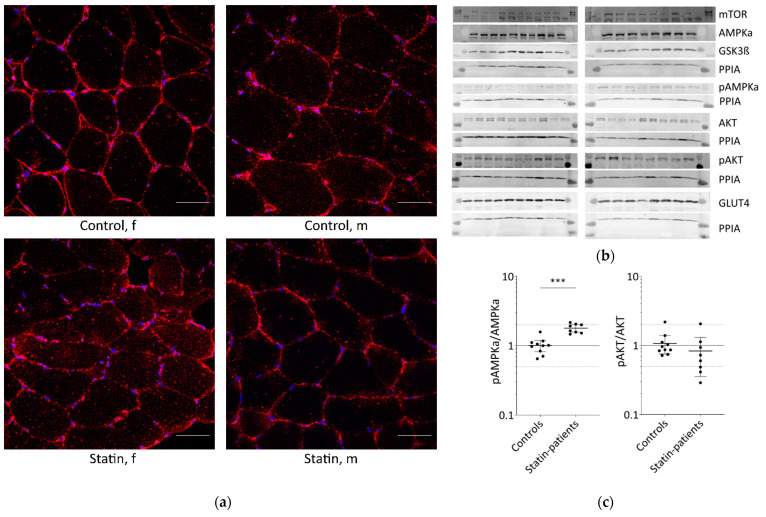
Proteins involved in glucose uptake and signaling in human skeletal muscle tissue from controls and statin-treated patients. (**a**). Immunohistochemistry staining for glucose transporter GLUT4 distribution. Images shown are representative data. (control female (C5), control male (C12), statin patient female (S1), statin patient male (S4)). (**b**). Western blot of human skeletal muscle tissue from statin-treated patients (right panel) and controls (left panel). Each lane corresponds to another sample (left panel = C1–C3, C5–C9, C11, and C12; right panel = S1–S8; see Table 1). PPIA is the reference protein. mTOR, AMPKα, and GSK3ß were probed on the same blot. Phosphorylated AMPKα, AKT, and phosphorylated AKT were analyzed on separate immunoblots. GLUT4 was probed on the same membrane as phosphorylated AMPKα. Signals shown are derived with an LI-COR-Odyssey infra-red scanner, except for GLUT4, and quantified using Image Studio Lite ver5.2. (**c**). Ratio of phosphorylated AMPKα and phosphorylated AKT to total AMPKα and AKT protein, respectively, to control for differences in total protein level. (Results obtained from controls C1–C3, C5–C9, C11, C12 and statin-treated patients S1–S8 (see Table 1); f = female; m = male; dotted line at 2- and 0.5-fold; *p* < 0.05, ** *p* < 0.005; *** *p* < 0.0005, Mann–Whitney test; plot shows single data points with median 95% confidence interval for the median). (**d**). Relative expression level of proteins relevant for glucose uptake regulation and metabolism in human muscle tissue.

**Figure 2 ijms-23-02398-f002:**
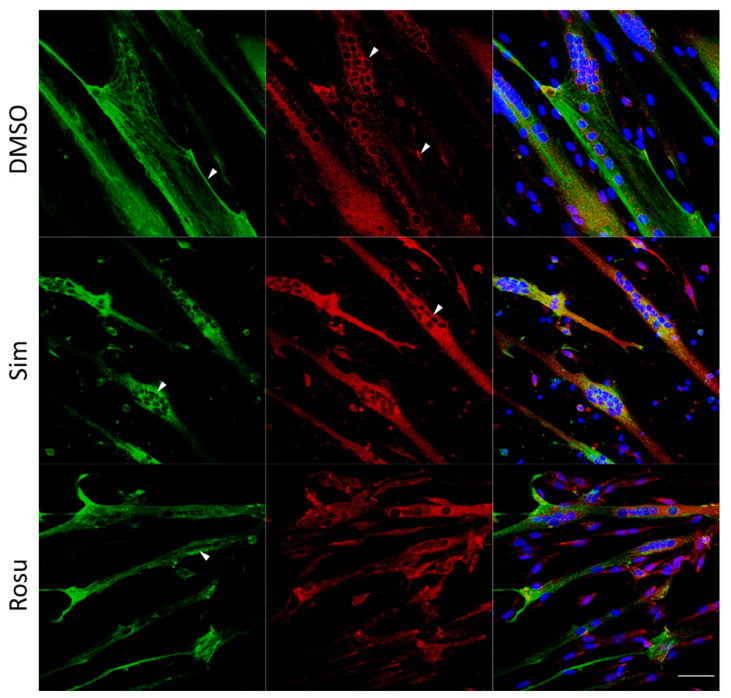
Caveolin 3 (CAV3) and glucose transporter GLUT4 immunofluorescence staining for non-stimulated statin-treated primary human myotubes and controls. Images shown are representative data. CAV3 (green) accumulates around the nucleus (arrow) and is less at the plasma membrane (arrow) after simvastatin treatment. GLUT4 (red) is normally distributed in vesicular structures around the nucleus (arrow) and at the plasma membrane (arrow). After simvastatin, GLUT4 signals are less punctuated (arrow) as expected for vesicular structures and observed for controls (arrow). Images shown are representative data. (Results obtained from C3–C5, C10–C12, and S8; see Table 1) UT = untreated; Sim = simvastatin; Rosu = rosuvastatin; scale bar = 50 µm.

**Figure 3 ijms-23-02398-f003:**
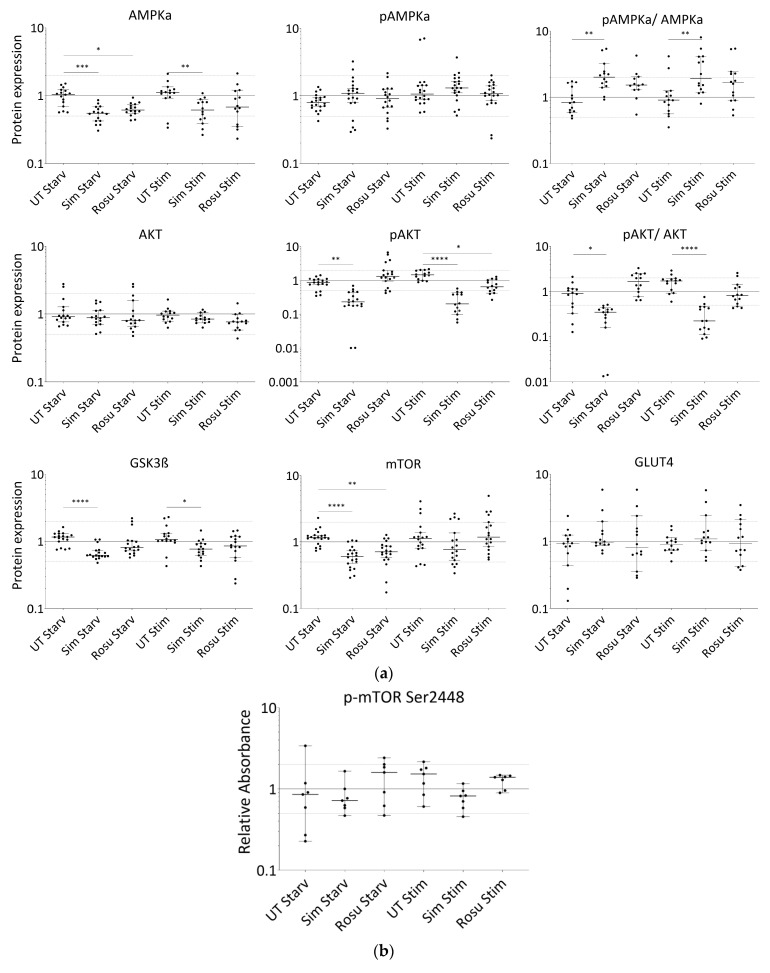
Expression of proteins relevant for glucose-uptake regulation and metabolism in human primary myotubes non-stimulated and stimulated with glucose and insulin under statin treatment. (**a**). Relative expression level of proteins relevant for glucose uptake regulation and metabolism under statin-treatment compared to its untreated control w/o glucose and insulin stimulation. Ratios of pAMPKα and pAKT to total AMPKα and AKT protein, respectively to control for differences in total protein levels. (**b**). mTOR phosphorylation at Ser2448 was detected using sandwich ELISA method. Values are protein level corrected. (**c**–**i**). Western blot of statin-treated human primary myotubes unstimulated and stimulated with insulin and glucose. PPIA and GAPDH were used as reference proteins. mTOR, AMPKα, and GSK3ß were probed on the same blot. Phosphorylated AMPKα, AKT, and phosphorylated AKT were analyzed on separate immunoblots. GLUT4 was probed on the same membrane as phosphorylated AMPKα. Signals shown are derived using an LI-COR-Odyssey infra-red scanner, except for GLUT4, and quantified via Image Studio Lite ver5.2. Protein ladder lanes separate each myotube cell line data set (C = C11; D = C5; E = C12; F = C4; G = C3; H = C10; I = S8; see Table 1). Each data set includes untreated, DMSO, simvastatin, rosuvastatin non-stimulated and stimulated with glucose and insulin, respectively. Western blotting was repeated twice. (Results obtained from C3–C5, C10–C12, and S8; see Table 1; GAPDH–bands derived with ECL method only) [UT = untreated; Sim = simvastatin; Rosu = rosuvastatin; Starv = starving condition only; Stim = insulin/glucose stimulation after starving; dotted line at 2- and 0.5-fold; * *p* < 0.05, ** *p* < 0.005; *** *p* < 0.0005, **** *p* < 0.00005; one-way ANOVA corrected for Dunn’s multiple comparison testing; plots show single data points with 95% confidence interval for the median].

**Table 1 ijms-23-02398-t001:** Clinical information and myopathological findings of patients used in this study. None of the patients have been diagnosed with T2DM. Controls were statin-naïve. All statin patients (S1–S8) were under statin treatment at the time of muscle biopsy. Unless otherwise stated, muscle biopsy specimens originated from M vastus lateralis. [§ used for in vitro experiments; # M triceps brachii].

Patient ID	Gender of Donor	Age of Donor	Medical History of Donor	Myopathological Findings in Biopsy Specimen
C1	f	47	No statins	Normal #
C2	f	54	No statins, myalgia of unknown origin, CK normal	Normal
C3 §	f	51	No statins, no myalgia, CK normal	Normal#
C4 §	f	59	No neuromuscular symptom, hip replacement surgery	Normal
C5 §	f	51	No statins, myalgia of unknown origin, CK normal	Normal; increased intracellular lipid droplets
C6	f	56	No statins, myalgia of unknown origin, CK normal	Normal, rare atrophic fibers
C7	m	65	No statins	Normal
C8	m	64	No statins	Normal
C9	m	66	No statins	Normal
C10 §	m	67	No neuromuscular symptom, hip replacement surgery	Normal
C11 §	m	50	No statins, myalgia of unknown origin, CK = 290 U/L	Normal; minor fiber type grouping #
C12 §	m	56	No statins, myalgia of unknown origin, CK = 513 U/L	Normal, rare central nuclei #
S1	f	58	Statin myopathy, Pain rate = 4 Simvastatin 20 mg; CK = 254 U/L	Normal
S2	f	60	Statin myopathy, Pain rate = 2 Simvastatin 20 mg; CK = 203 U/L	Normal
S3	m	71	Statin myopathy, Pain rate = 2 Atorvastatin 10 mg; CK = 209 U/L	Normal
S4	m	64	Statin myopathy, Pain rate = 3 Atorvastatin 10 mg; CK = 527 U/L	Normal
S5	m	62	Statin myopathy, Pain rate = 4 Pravastatin 10 mg; CK = 688 U/L	Normal
S6	m	58	Statin myopathy, Pain rate = 5 Atorvastatin 40 mg; CK= 430 U/L	Normal
S7	m	53	Statin myopathy, Pain rate = 3 Pravastatin 5 mg; CK = 534 U/L	Normal
S8 §	m	77	Statin patient without side effects; Simvastatin 10 mg; CK = 195 U/L	Normal

**Table 2 ijms-23-02398-t002:** Antibodies and dilutions for Western blotting and immunofluorescence.

Antibody	Supplier	Dilution
Akt (#9272)	Cell Signaling Technology (Beverly, MA, USA)	1:2000
phosphorylated Akt (Ser473; #9271)		1:1000
AMPKα (#2603)		1:1000
phosphorylated AMPKα (Thr172; #2535)		1:1000
GSK3β (27C10)		1:1000
mTOR (#2983)		1:1000
Cyclophilin A/ PPIA (#ab41684)	Abcam (Cambridge, UK)	1:5000
GAPDH (#ab9484)		1:10000
GLUT4 (#ab654); used in immunofluorescence staining and Western blotting with primary human myotubes		1:3000
GLUT4 antiserum (1154 p); used in Western blot and immunohistochemical muscle tissue staining with human skeletal muscle tissue	kindly provided by Hoffmann-La Roche (Nutley, NJ, USA) to Hadi Al-Hasani and AG Schuermann (DIFE Potsdam, Nuthetal, Germany)	1:1000
CAV3 (sc-5310)	Santa Cruz (CA USA)	1:150
Alexa 488 and Alexa 568	Invitrogen (Berlin, Germany)	1:500
Biotin goat anti-rabbit ab and Cy3-conjugated streptavidin	Jackson ImmunoResearch Laboratories (West Grove, PA, USA)	1:200
ECL™ IgG, HRP linked	GE-Healthcare (Waukesha, WI , USA)	1:2000
IRDye 800	Rockland (Washington, DC, USA)	1:5000

## Data Availability

Primary Western blot data are included in the manuscript. All other primary data are available upon science-based request via figshare.

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
