# Peer review of "Statins Aggravate the Risk of Insulin Resistance in Human Muscle"

_ijms, 2022, doi:10.3390/ijms23042398_

Round 1
Reviewer 1 Report
The manuscript ‘ Statins aggravate the risk of insulin resistance in human muscle’ provides the new insight into statin treatment in terms of insulin resistance inhuman muscles. This is an interesting field with much activity over the last 10-15 years.
The text is well organized and the findings could have significant implications given the fact that statins are the most prescribed drugs for treating cardiovascular disease. Although the results are of interest, I have the following concerns:
- The authors show that GLUT4 protein expression levels were not statistically significantly different between statin treatment and controls. However, previously it was demonstrated that after statintreatment GLUT4 expression in HSkMCs was suppressed (DOI: 2147/DDDT.S87979).
Please commend this opposite results
- Akt phosphorylates TBC1D1 and TBC1D4 reducing the inactivation of Rab GTPases that confer directionality to GLUT4 translocation (DOI: 1021/acs.jmedchem.1c00410). How the authors interpret their results under condition of statin treatment and subsequent loss of isoprenoid metabolites necessary for Rab prenylation and activation.
- type-2 diabetes and type 2 diabetes mellitus are used
Author Response
Dear Reviewer,
many thanks for taking the time to review our manuscript. We also thank you for the valuable comments, especially on the interesting publications, the inclusion of which makes our manuscript better and more relevant.
We also would like to respond to your comments.
Point (1) "The authors show that GLUT4 protein expression levels were not statistically significantly different between statin treatment and controls. However, previously it was demonstrated that after statintreatment GLUT4 expression in HSkMCs was suppressed (DOI: 2147/DDDT.S87979).
Please commend this opposite results."
Response to 1:
We have read the publication of carefully and added it to the discussion.
Both, insulin stimulation via AKT and muscle contraction via AMPK regulate GLUT4 translocation and GLUT4 expression, thereby modulating glucose homeostasis in skeletal muscle.
As we did find decreased activation of AKT, AMPK activity was increased leading to GLUT4 translocation to cell membrane and GLUT4 expression levels comparable to controls. Another study perfomed by Zhao and Zhao (2015) also showed decreased p-AKT but contrary to our results found GLUT4 expression reduced accompanied by p-p38 MAPK expression being reduced, which is downstream of AMPK and explains the decrease in GLUT4 expression found. Diversities in the donor and statin concentration used may explain the difference between the studies. However, activated AMPK is increased and thus GLUT4 expression is not significantly decreased in our study comprising results from 20 different donors with individual variances being found. GLUT4 expression levels were confirmed at protein level by immunohistochemistry and Western blot in tissues from statin patients and in the statin-treated muscle cells. (1,2)
Additionally, we would like to mention that the authors, understandably due to the difficulty of accessing the material, purchased the HKMCs, with the known difficulties associated with having little information about the donor (e.g., what pre-existing conditions and medications the donor was taking at the time the skeletal muscle sample was collected). Since a university outpatient clinic is connected to our laboratories, we do have all the information and we have taken into account previous diseases and medication. As I am sure you have seen, individual differences occur between donors that would explain the differences from our data. The level of phosphorylated AKT is decreased in the study by Zhao and Zhao as in our work. A measurement of AMPK was not performed by the authors, but p38 MAPK was. However, it is known that activation of AMPK subsequently phosphorylates p38 MAPK and the authors determined the phosphorylated form of p38 MAPK to be decreased under certain statin treatment. This may indicate decreased AMPK activation, which explains the authors' decreased GLUT4 expression. Activated AMPK is increased in our results and thus GLUT4 expression is not decreased. We confirmed our GLUT4 expression levels at protein level by immunohistochemistry and Western blot in tissues from statin patients and in the statin-treated muscle cells.
Point (2) "Akt phosphorylates TBC1D1 and TBC1D4 reducing the inactivation of Rab GTPases that confer directionality to GLUT4 translocation (DOI: 1021/acs.jmedchem.1c00410). How the authors interpret their results under condition of statin treatment and subsequent loss of isoprenoid metabolites necessary for Rab prenylation and activation."
Response to 2:
This is an interesting point and we addressed that question in the discussion.
In different cell types, such as adipocytes and ß-islet cells, statins are described to have a positive effect on insulin sensitivity and hyperglycemia. However, this is different for muscle, which may switch to an alternative pathway for GLUT4 translocation. (1) Statins inhibit the formation of isoprenoids and thus farnesyl and geranylgeranyl pyrophosphates, coming from the mevalonate pathway and required for small GTPase prenylation, are reduced. (1,3) RabGTPases are hydrophobically prenylated by Rab geranylgeranyl transferase to bind to their target membrane structures, such as GLUT4 vesicles, and to direct the translocation of GLUT4. (4,5) Furthermore, prenylation of Ras and Rab leads to phosphorylation of PI3K, which activates Akt and mTOR, which is decreased in our study. However, activation of AKT leads to phosphorylation and inactivation of TBC1D1 and TBC1D4. The activity of TBC1D1 and TBC1D4 is thought to inhibit the activity of Rab GTPase-activating protein toward certain Rab isoforms. Inactivation of TBC1D1 and TBC1D4 results in the increased generation of more active GTP-loaded Rab. Active GTP-loaded Rab promotes GLUT4 translocation to the plasma membrane. Active AMPK can also mediate the emergence of more active GTP-loaded Rab. (1,3,4) Prenylation of small GTPases may be depleted over time by inhibition of the mevalonate pathway and even by an increase in HMGCR expression, as we showed in Grunwald et al 2020 under statin treatment. The results from muscle tissue samples from patients treated with statins for years show normal GLUT4 translocation. None of our patients had type 2 diabetes. It would be interesting to study how this changes should the patients develop type 2 diabetes.
Point (3) "type-2 diabetes and type 2 diabetes mellitus are used"
Response to 3:
Thank you for pointing this out. We changed that.
We hope that we have addressed your comments sufficiently and wish you success in your work.
Kind regards
Simone Spuler and Stefanie Grunwald, on behalf of all authors of the study
References
(1) Edyta Gendaszewska-Darmach, Malgorzata A. Garstka, and Katarzyna M. Błażewska.
Targeting Small GTPases and Their Prenylation in Diabetes Mellitus. (2021) Journal of Medicinal Chemistry 2021 64 (14), 9677-9710 DOI: 10.1021/acs.jmedchem.1c00410
(2) Zhao W, Zhao SP. Different effects of statins on induction of diabetes mellitus: an experimental study. Drug Des Devel Ther. 2015;9:6211-6223. Published 2015 Nov 24. doi:10.2147/DDDT.S87979
(3) Hyder T, Marti JLG, Nasrazadani A, Brufsky AM. Statins and endocrine resistance in breast cancer. Cancer Drug Resist 2021;4:356-64. http://dx.doi.org/10.20517/cdr.2020.112
(4) Kei Sakamoto and Geoffrey D. Holman. Emerging role for AS160/TBC1D4 and TBC1D1 in the regulation of GLUT4 traffic. (2008) Am J Physiol Endocrinol Metab 295: E29 –E37, 200
(5) Kirill Alexandrov, Yaowen Wu, Wulf Blankenfeldt, Herbert Waldmann, Roger S. Goody. 8 - Organization and Function of the Rab Prenylation and Recycling Machinery. (2011) The Enzymes. Academic Press. Vol 29; 147-162. https://doi.org/10.1016/B978-0-12-381339-8.00008-1.
Reviewer 2 Report
Statins are now well documented for inducing glucose intolerance. The effect has been tested both in animal models and clinical studies.
The pathology accounted for multiple disrupted signaling. This study tested the long-term and short-term effects of statins on skeletal muscles. Now, when metabolic disorders are on the rise and one of the most popular drug treatments influencing another metabolic disorder is a matter of concern. Therefore, the study is time-relevant indeed.
The overall study design is satisfactory. However, the figures need significant improvement in the arrangement and presentation. In figure 1, the numbering convention is tough to follow. Please place the marking letters in a sequence and a specific position of the corresponding panel (left upper corner). The immunohistochemistry image panel could be the Fig 1A, and details should be below the respective image. Each histological image should have a scale bar. Please present the immunoblots in a legible size in the final published version.
Please mention in the data if there is any influence of the sex of the donor?
Author Response
Dear Reviewer,
Many thanks for taking the time to review our manuscript. We also thank you for the valuable comments, especially on the interesting publications, the inclusion of which makes our manuscript better and more relevant.
We also would like to respond to your comments.
Point (1) "In figure 1, the numbering convention is tough to follow. Please place the marking letters in a sequence and a specific position of the corresponding panel (left upper corner). The immunohistochemistry image panel could be the Fig 1A, and details should be below the respective image. Each histological image should have a scale bar. "
Response to 1:
We very much agree with you that Figure 1 is difficult to follow. To create this image panel, we followed the guidelines of the template from the Journal. According to your suggestion, we have adapted Figure 1 and hope that this is still in line with editor´s standards.
Point (2) Please present the immunoblots in a legible size in the final published version.
Response to 2:
We increased the size of immunoblots, in particular in Figure 3.
Point (3) Please mention in the data if there is any influence of the sex of the donor?
Response to 3:
Thank you very much for this good question. We have indeed checked this in our analyses and found no relevant difference between the genders.
We hope that we have addressed your comments sufficiently and wish you success in your work.
Kind regards
Simone Spuler and Stefanie Grunwald, on behalf of all authors of the study
This manuscript is a resubmission of an earlier submission. The following is a list of the peer review reports and author responses from that submission.